# Protective effect of luteinizing hormone on frozen-thawed ovarian follicles and granulosa cells

Jie Chen[1,2] *, Boyang Yu[2], Shengbo Zhang[1], Zhikang Wang[1], Yanfeng Dai[1] *

1 School of Life Science, Inner Mongolia University, Hohhot, PR China, 2 College of Basic Medicine, Inner Mongolia Medical University, Hohhot, PR China

⊕ These authors contributed equally to this work.
* chenjie@immu.edu.cn (JC); daiyf@imu.edu.cn (YD)

**Data Availability Statement:** All relevant data are within the manuscript and its Supporting Information files.

**Funding:** This study was funded by the Natural and Science Foundation of Inner Mongolia

## Abstract

Ovarian tissue cryopreservation addresses critical challenges in fertility preservation for pre-pubertal female cancer patients, such as the lack of viable eggs and hormonal deficiencies. However, mitigating follicle and granulosa cell damage during freeze-thaw cycles remains an urgent issue. Luteinizing hormone (LH), upon binding to luteinizing hormone receptors (LHR) on granulosa cells, enhances estrogen synthesis and secretion, contributing to the growth of granulosa cells and follicles. This study examined mouse ovarian follicles and granulosa cells to identify optimal LH treatments using morphological assessments and LIVE/DEAD assays. The study found significant increases in the expression of Leucine-rich G-protein-coupled receptor 5 (Lgr5) and Forkhead box L2 (Foxl2) in mural and cumulus granulosa cells under LH influence, alongside marked reductions in active caspase-3 expression. Double immunofluorescence of Ki67 with Foxl2 and Lgr5 revealed ongoing pro-liferative activity in granulosa cells post freeze-thaw. In addition, LH treatment significantly boosted the expression of transforming growth factor (TGF-β) and its superfamily members in both granulosa cells and oocytes. These findings suggest that LH addition during cryo-preservation can diminish damage to follicles and granulosa cells, offering new strategies to enhance the efficacy of mammalian ovarian cryopreservation.

## 1. Introduction

Vitrification cryopreservation employ a high concentration of cryoprotectants to achieve rapid freezing and reduce ice crystal formation. This method significantly reduces the freezing damage associated with slow freezing, preserving the biological functiona and inherent activity of cells [1]. For young female cancer patients unable to harvest viable follicles for cryopreservation, vitrification of ovarian tissue prior to cancer treatment represents a feasible strategy for fertility preservation. This approach allows for the restoration of the ovary's natural rhythm post-radiotherapy and chemotherapy following replantation [2]. Preliminary research indicates that the protective effect of LH surpasses that of follicle-stimulating hormone (FSH) during vitrification cryopreservation [3]. Granulosa cell apoptosis is a leading cause of ovarian

Autonomous Region (No. 2019MS08139, 2022QN03003). Role of Funder statement: Jie Chen: Conceptualization, Formal Analysis, Funding acquisition, Visualization, Writing - Original Draft, Writing - Review & Editing. Boyang Yu: Conceptualization, Funding acquisition, Investigation, Validation, Writing - Review & Editing.

**Competing interests:** The authors have declared that no competing interests exist.

follicular atresia in mice [4]. LH binds to its receptor on granulosa cells during the intermediate and late stages of follicular development, initiating signaling pathways that induce specific protein expression [5]. Granulosa cells consist of two groups of pregranulosa cells that develop successively. The first group expresses Foxl2 positive cells in embryonic ovaries during sex determination after mating in mice [6]. The second group originates from Lgr5 positive cells in the ovarian epithelium, which determine the female follicular pool and sustain female reproductive functionality over an extended duration [7, 8]. Foxl2 exhibits stable expression in fetal and postnatal granulosa cells [9]. As a unique granulosa cells marker, its upregulation can effectively ensure the paracrine signaling of granulosa cells, maintain the stability of the follicular membrane [10]. Lgr5-positive cells eventually transition to Foxl2-positive cells, facilitating the formation of primitive follicles by enveloping oocytes [11]. After the establishment of the initial ovarian follicle pool, only the ovarian epithelium exhibits Lgr5 expression, which is critical for epithelial repair following ovulation. Foxl2 expression in granulosa cells facilitates their growth and differentiation [12]. Researchers have leveraged the expression patterns of these granulosa cell-specific proteins, namely Foxl2 and Lgr5, to determine the composition and abundance of these cells [13]. The presence of LHR on granulosa cell membranes indicates their capacity to bind LH, reflecting their functional status after freeze-thaw cycles. This observation provides conclusive evidence of the role of LH in enhancing the recuperation of granulosa cells from freeze-thaw damage and identifies its precise targets of action.

The prevailing view held that apoptosis was the only regulated mode of cell death, essential for maintaining homeostasis and influencing cellular development [14]. Apoptotic cells exhibit morphological changes such as shrinkage and are governed by key signaling pathways, namely, exogenous and endogenous apoptosis signaling, along with caspase-independent pathways [15]. Activation of caspase-3 leads to genomic instability and cell proliferation. Therefore, implementing strategies to inhibit apoptosis is crucial, particularly for protecting freeze-thaw granulosa cells. The caspase protease family plays a central role as the primary effector in the apoptosis pathway, requiring activation by both internal and external signals to trigger the apoptotic process [14]. caspase-3, integral to the protease cascade of apoptosis, triggers extensive granulosa cell apoptosis and marks an irreversible phase of apoptosis upon its activation [16]. This study focuses on activated caspase-3, which is primarily involved in executing apoptosis and initiating DNA degradation [17].

Maturation of follicles in vivo is governed by a complex network of signals [18]. Of particular note, the TGF-β superfamily plays a critical role in this process through its regulation of intracellular signaling [19]. Expressed within oocytes, Bone morphogenetic protein-15 (BMP-15) and growth differentiation factor-9 (GDF-9), both members of the TGF-β superfamily, enhance granulosa cell proliferation and differentiation via signaling through the TGF-β pathway [20–22]. Research indicates that the deletion of BMP subgroups in murine ovarian follicles can lead to severe consequences, including impaired fertility [23], highlighting the critical role of TGF-β superfamily members in female reproductive health. Structural variations in the GDF-9 gene, linked to granulosa cell development, have been detected in patients with premature ovarian insufficiency [24]. Despite these variations, follicular counts remain normal, indicating a strong link between GDF-9 and the initiation of granulosa cell development. As part of the TGF-β superfamily, GDF-9 influences follicular maturation and activates signaling pathways in cumulus granulosa cells, enhancing oocyte development and maturation while also controlling granulosa cell differentiation and proliferation [25, 26].

While the protective influence of LH in vitrification cryopreservation of ovaries is well-documented, the direct effects of LH on the cryopreservation of follicles and granulosa cells, as well as the underlying mechanisms, are crucial for enhancing the efficiency of cryopreservation and advancing the technology of female mammalian gonad cryopreservation.

## 2. Materials and methods

DOI: dx.doi.org/10.17504/protocols.io.bp2l62pedgqe/v1 (Private link for reviewers: https://www.protocols.io/private/45D3D3115ABA11EFB7510A58A9FEAC02 to be removed before publication.)

### 2.1 Ethics statement

Approval for all experimental protocols was obtained from the Committee for Ethics on Animal Care and Experiments at Inner Mongolia University. Euthanasia was conducted using carbon dioxide to ensure minimal animal distress, with all protocols designed to reduce suffering to the greatest extent possible.

### 2.2 Animals and reagents

**2.2.1 Animals.**   Female C57BL/6J mice, aged four weeks and maintaining SPF conditions, were acquired from the Laboratory Animal Center at Inner Mongolia University which holds Experimental Animal License Number SYXK (Mongolia) 2020–0006. These mice were housed under SPF conditions, with ad libitum access to food and water. Environmental controls were set to maintain temperatures between 20 and 24°C, a relative humidity of 60%, and natural light cycles were preserved.

**2.2.2 Principal reagent LH.**   Recombinant human LH for injection, with a purity of 100% (Merck Serono, DM). Building on the concentration gradient established in our prior studies, an intervention concentration of 0.3 IU/mL LH was selected [3].

### 2.3 Experimental grouping and protocol

**2.3.1 Grouping.**   In the study examining optimal LH intervention protocols, eighty ovaries were randomly divided into four groups of twenty each. Grouping was as follows: ① Control group: Fresh ovarian tissue was fixed immediately after collection. ② LH-BV group: 0.3 IU/mL LH was introduced prior to cryopreservation.③LH-AV group: 0.3 IU/mL LH was added post-cryopreservation.④LH-TV group: 0.3IU/mL LH was administered throughout the cryopreservation process. After freeze-thaw, tissue samples from each group were subjected to morphological assessment and cell counting, while another set was analyzed for viability of granular cells using LIVE/DEAD staining.

**2.3.2 Protocol.**   In an experimental study examining the effects of LH interventions, sixty ovaries were randomly divided into three groups of twenty each. Group assignments were as follows: ①Control group: no vitrification or cryopreservation post-ovary sampling. ②Vitrification group: ovarian tissue underwent vitrification cryopreservation without LH intervention. ③LH group: ovarian tissue was treated with the full vitrification cryopreservation process, incorporating 0.3IU/mL LH throughout. After the group processing is complete, several analytical techniques were applied. Granulosa cells were isolated using a stereomicroscope for morphological assessment and quantification. Tissues post-freeze and thaw were fixed in 4% paraformaldehyde (PFA) for 24 hours, followed by preparation of paraffin sections. Both immunohistochemistry and immunofluorescence analyses were performed. Total RNA was extracted for real-time PCR, and total protein was analyzed via Western blot.

### 2.4 Vitrification-based cryopreservation procedures

The protocol involved a preliminary incubation at 37°C in a 5% $CO_2$ environment for a 60 minutes, followed by an initial equilibrium phase at the same temperature for 8 minutes. The osmotic equilibrium stage employed a vitrification solution at 37°C for 4 minutes. The

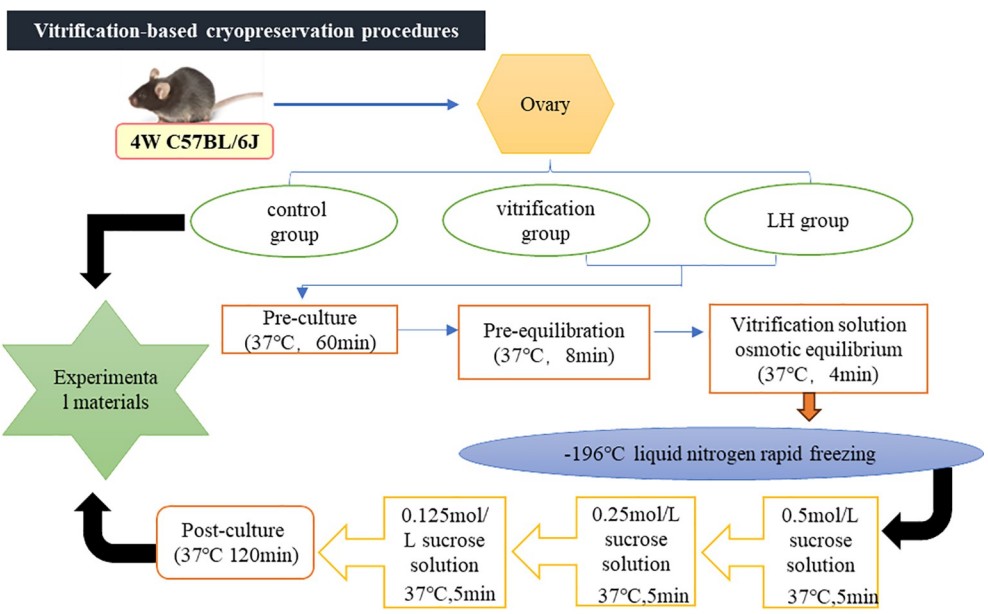

**Fig 1. Vitrification-based cryopreservation procedures.**

equilibrated ovarian tissue was quickly transferred to liquid nitrogen for overnight cryogenic freezing. Upon retrieval from the liquid nitrogen, the frozen tissue was immersed in a decreasing concentration gradient of thawing solution to facilitate the removal of any residual cryopreservation solution. A post-culture phase at 37°C in a 5% $CO_2$ environment for 120 minutes completed the vitrification cryopreservation process (Zheng X et al.,2020) (Fig 1).

## 2.5 Morphological observation and follicle counts

To prepare paraffin sections, account for the varied sizes of follicles at different developmental stages to prevent counting the same follicle multiple times. One section every five slices was selected, yielding six sections for HE staining. For each section, five high-power fields were examined under a microscope to assess follicle morphology. The viable follicles at each developmental stage were counted. Follicles characterized by shrinkage of the oocyte nucleus and disordered granulosa cell arrangement were identified as apoptotic. Finally, the percentage of viable follicles at each stage of development was calculated.

## 2.6 LIVE/DEAD detection

The experiment employed a Live/Dead cell staining kit (Calcein AM, PI method) sourced from Proteintech, Wuhan, Chian. The process involved thawing the necessary reagents, followed by vigorous vortexing to mix 30 μL of 1.5 mM PI with 5 μL of 4mM Calcein AM with in a 10 mL solution of PBS, obtained from Seven, Beijing, China. Then the cellular samples underwent a 30-minute incubation at ambient temperature within this mixture. Following the incubation, the cells were rinsed with 1 x PBS, after which imaging and quantification were performed to assess the viability of the granulosa cells. Each sample contained a cell density of 1 x $10^6$/mL.

## 2.7 Detection of differentiation and proliferation by immunofluorescence double labeling in vitro

To complete dehydration, the sections underwent sequential immersions in a series of ethanol solutions of diminishing concentrations: 100%, 95% and 70% at ambient conditions. EDTA8.0

(Servicebio, China) facilitated antigen retrieval; following this, the sections were exposed to a 3% hydrogen peroxide solution for a duration of 25 minutes to block endogenous peroxidase activity. A 30-minute block was then achieved using 3% BSA. Sections were treated with a mouse-derived monoclonal antibody against Ki67 (1:200, Servicebiotech China) and maintained at 4°C overnight. Post three PBS washes, incubation proceeded with HRP-conjugated goat anti-mouse IgG for 50 minutes at ambient conditions. TSA exposure lasted for 10 minutes at the same temperature. A second round of antigen retrieval preceded the addition of either rabbit-derived monoclonal antibodies against Foxl2 (1:200, abcam, UK) or Lgr5 (1:1000, abcam, UK), which were also incubated overnight at 4°C. Following another trio of PBS washes, sections were further incubated with HRP-conjugated goat anti-rabbit IgG for 50 minutes at room temperature. Nuclei restaining was then performed, culminating in microscopic data analysis.

## 2.8 Granulosa cells were cultured in vitro

Cumulus cell complexes (COCs) were isolated from all ovaries in each experimental group and rapidly separated. All follicles from each group were then introduced into culture droplets. Using a 1 mL syringe needle, the follicular membrane was incised to release the granulosa cells, which were transferred to 6-well plates for overnight culturing. Each well contained approximately $1x10^6$ cells.

## 2.9 Granulosa cell count

The suspended granular cells were harvested from the culture medium of in vitro cultured granular cells. Adherent granular cells were treated with 0.25% pancreatic enzyme for digestion and then centrifuged at 1200 rpm for 5 minutes. The pellet was resuspended in 1 mL of 1xPBS, and a 15 μL aliquot of the single-cell suspension was used for cell counting using an automatic cell counter (BODBOGE, Guangzhou, China). Cell counts were performed in triplicate, and the average was calculated for statistical analysis.

## 2.10 Immunohistochemical detection

Ovarian tissues were subjected to freeze-thaw cycles according to specified groupings and subsequently fixed in 4% PFA for a period of 24 hours. Following fixation, the tissues underwent paraffin embedding to facilitate the creation of sections. These sections were then subjected to a dehydration process using ethanol concentrations that gradually decreased, performed at ambient temperature. Antigen retrieval was conducted in a 0.01M citric acid buffer for 8 minutes using microwave assistance. To block endogenous peroxidases and nonspecific binding, sections were treated with 3% hydrogen peroxide for 10 minutes and 10% goat serum for 30 minutes, respectively. They were then incubated with primary antibodies overnight at 4°C, including rabbit monoclonal anti-Foxl2 at a 1:2000 dilution, rabbit polyclonal anti-Lgr5 at a 1:400 dilution, rabbit polyclonal anti-active caspase-3 at a 1:200 dilution, rabbit monoclonal anti-TGF-β at a 1:500 dilution, rabbit polyclonal anti-GDF-9 at a 1:500 dilution, and rabbit polyclonal anti-BMP-15 at a 1:200 dilution (all antibodies from abcam, UK). Following three PBS washes, the sections were exposed to HRP-conjugated goat anti-rabbit IgG for 1 hour at room temperature. After rinsing with PBS, the sections were developed using DAB (ZSGB-BIO, Beijing, China), with the reaction monitored microscopically and halted promptly. Control experiments used PBS instead of primary antibody for the negative controls.

## 2.11 Immunofluorescence detection

Paraffin-embedded tissue sections were first dehydrated and subjected to antigen retrieval followed by blocking with serum. These sections were incubated at 4˚C overnight using primary antibodies, specifically Foxl2 and Lgr5 for immunohistochemistry, as well as a rabbit polyclonal antibody to LHR at a dilution of 1:200 (abcam, UK). In the second day, these sections were brought to room temperature over an hour before being treated with a secondary antibody, Goat Anti Rabbit IgG at a dilution of 1:200 (proteintech, China), and incubated for 1 hour at 37˚C in an environment protected from light. After a 10-minute staining of cell nuclei with DAPI, the sections were sealed and microscopic examination was conducted to capture images.

## 2.12 Quantitative analysis of protein imprinting

Total protein extraction was carried out using the ProteinExt Mammalian Total Protein Extraction Kit from TransGen Biotech, Beijing, China. To standardize the protein concentrations across samples, protein quantification was performed with a BCA kit from the same manufacturer. Each sample, integrated with 1 x loading buffer, was heated at 95˚C for five minutes to denature the proteins. Based on molecular weights, proteins were subsequently separated using sodium dodecyl sulfate-polyacrylamide gel electrophoresis (SDS-PAGE) at varying concentrations. 12% SDS-PAGE was utilized for the separation of proteins tagged with rabbit monoclonal antibodies against Foxl2/TGF-β and rabbit polyclonal antibodies against GDF-9/ BMP-15 (1:1000 dilution, abcam, UK). An 8% SDS-PAGE was used for rabbit polyclonal antibodies against Lgr5 (1:1000, abcam, UK), and a 15% SDS-PAGE was employed for rabbit polyclonal antibodies against active caspase-3 (1:2000 dilution, abcam, UK). The resolved protein bands were transferred onto polyvinylidene difluoride (PVDF) membranes, which were subsequently blocked with a 5% skim milk solution at room temperature for one hour. The PVDF membranes were then incubated overnight at 4˚C with primary antibodies. After three TBST washes, the membranes were exposed to HRP-conjugated goat anti-rabbit IgG secondary antibodies at 37˚C for one hour. Detection of the target proteins was achieved using the ultra-sensitive ECL chemiluminescence kit from Proteintech, Wuhan, China.

## 2.13 Total RNA extraction

Total RNA was isolated from 20 ovaries per experimental group using Trizol (TAKALA, 9108, RNAiso Plus, JPN). Subsequent cDNA synthesis was carried out with the TransStart One-Step gDNA Removal and cDNA Synthesis SuperMix kit (TransGen, Beijing, China). Adhering to the protocol provided by the kit and based on RNA concentration, 1 μg of RNA was utilized for the reverse transcription process.

## 2.14 Real-time PCR

The synthesis of cDNA involved primer sets listed in Table 1, provided by Shanghai Shenggong. For the PCR amplifications, TB Green premix Ex Taq II (Tli RnaseH Plus) (TAKALA, RR820, JPN) was used. The protocol for amplification included an initial denaturation at 95˚C for 2 minutes, followed by 40 cycles of denaturation at 95˚C for 5 seconds, annealing at 58˚C for 30s seconds, and extension at 72˚C for 30 seconds. This was concluded with a final extension phase at 72˚C for 10 minutes.

## 2.15 Statistical analysis

Statistical analysis of the data was performed using SPSS20.0, where quantitative data were presented as mean ± standard deviation (mean±SD). The LSD test within the framework of one-

**Table 1. Primer sequences used for real-time quantitative PCR.**

| Target gene | GenBank accession | Primer sequence | product size (bp) | Annealing temperature(°C) |
|---|---|---|---|---|
| Foxl2 | NM_012020 | F:ACAACACCGGAGAAACCAGAC<br>R:CGTAGAACGGGAACTTGGCTA | 145 | 62 |
| Lgr5 | NM_010195 | F:CCTACTCGAAGACTTACCCAGT<br>R:GCATTGGGGTGAATGATAGCA | 165 | 60 |
| Active caspase-3 | NM_00009810 | F:ATGGAGAACAACAAAACCTCAGT<br>R: TTGCTCCCATGTATGGTCTTTAC | 74 | 60 |
| TGF-β | NM_000660 | F:GGCCAGATCCTGTCCAAGC<br>R:GTGGGTTTCCACCATTAGCAC | 201 | 62 |
| GDF-9 | NM_008110 | F: TCTTAGTAGCCTTAGCTCTCAGG<br>R: TGTCAGTCCCATCTACAGGCA | 116 | 62 |
| BMP-15 | NM_009757 | F: TCCTTGCTGACGACCCTACAT<br>R: TACCTCAGGGGATAGCCTTGG | 100 | 62 |
| LHR | NM-018532 | F: CGCCCGACTATCTCTCACCTA<br>R:GACAGATTGAGGAGGTTGTCAAA | 150 | 62.3 |
| GAPDH | NM_008084 | F: AGGTCGGTGTGAACGGATTTG<br>R: TGTAGACCATGTAGTTGAGGTCA | 123 | 62 |

way ANOVA was employed to compare metrics across different groups. Statistical significance was acknowledged when the P-value was below 0.05.

## 3. Result

### 3.1 Morphological observation of LH intervention in different stages of vitrification cryopreservation

Histological evaluation using HE staining demonstrated a marked similarity in follicle morphology and structure between the LH-TV group and the control, suggesting that LH effectively protected against damage during the freeze-thaw process. On the contrary, the LH-BV and LH-AV groups exhibited significant follicular atresia and a disorganized granulosa cell layer compared to the LH-TV group (Fig 2A). The viability of granulosa cells post freeze-thaw was assessed using LIVE/Dead staining; results indicated that live cells comprised 66% of the control group and 75% in the LH-TV group, whereas only 45% and 38% of cells were viable in the LH-BV and LH-AV groups, respectively (Fig 2B–2F).

Follicular counts at various developmental stages were assessed across the four groups to evaluate the efficacy of protection in freeze-thawed follicles. In terms of primordial and primary follicles, the control group exhibited the highest number of viable structures, statistically significant with a P-value less than 0.05. For secondary and mature follicles, viability ranked from the LH group to the LH-BV and LH-AV groups, with the control group having the least. The combined count of viable follicles was notably greater in the control and LH-TV groups compared to the LH-BV and LH-AV groups, also with statistical significance marked by a P-value less than 0.05. Despite these variations, the overall follicle counts were comparable across all groups as shown in Table 2.

### 3.2 Results of granulosa cell viability detection and counting analysis

Ki67, a specific marker for cell proliferation, predominates in the nuclei of proliferative cells and is detected concurrently with Foxl2 and Lgr5, thereby serving as an indicator of granulosa cell vitality in thawed ovarian tissues. Quantitative analysis was performed on the dual expression of green fluorescent protein, tagged with Ki67, and red fluorescent protein, tagged with Foxl2 and Lgr5. Analysis of Ki67 and Foxl2 co-labeled cells revealed a higher frequency of

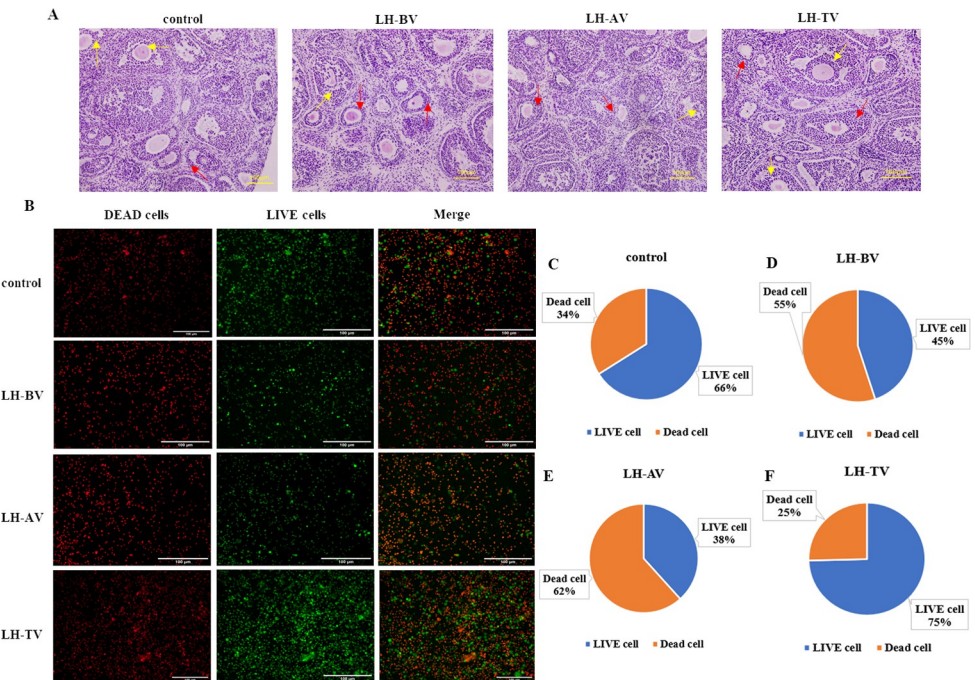

**Fig 2. Morphological analysis of LH addition at different periods of vitrification cryopreservation.** (A) HE staining results of LH intervention at different stages of freeze thawing. The red arrow represented parietal granulosa cells, and the yellow arrow represented cumulus granulosa cells, scale = 100μm. (B) LIVE/DEAD test results of 4 groups, scale = 100μm. (C) Pie chart of the proportion of living cells and dead cells in control group. (D) Pie chart of the proportion of living cells and dead cells in LH-BV group. (E) Pie chart of the proportion of living cells and dead cells in LH-AV group. (F) Pie chart of the proportion of living cells and dead cells in LH-TV group.

dual-fluorescent cells in the LH group compared to a markedly lower count in the vitrification group, with statistical significance noted (*P*<0.05) (Fig 3A and 3B). Assessment of Ki67 and Lgr5 co-expression identified the maximum number of cells displaying both red and green fluorescence in the LH group, in contrast to the minimal count observed in the vitrification group, also significant at *P*<0.05 (Fig 3C and 3D). In vitro culture of granulosa cells demonstrated that the LH group preserved cell morphology and structure more effectively than both the control and the vitrification groups (Fig 3E). Cell viability assessments post-thaw revealed the highest number of granulosa cells per high power field in the LH group, with the control group following and the vitrification group having the least. The vitrification group also showed the highest number of nonviable cells, while the LH group had the highest total cell count, significant at *P*<0.05 (Table 3).

**Table 2. The number and percentage of viable follicles at different developmental stages in four groups (mean ± SD) (n = 20).**

| Group | Number of viable original follicles | Number of viable primary follicles | Number of viable secondary follicles | Number of viable mature follicles | Total number of viable follicles at all levels | Total number of follicles at all levels | Percentage of viable follicles (%) |
|---|---|---|---|---|---|---|---|
| control | 28.36±0.25 | 31.52±0.22 | 11.08±0.33 | 18.34±0.28 | 89.3±0.22 | 97.23±0.31 | 91.84±1.01 |
| LH-BV | 4.75±0.11[a] | 14.63±0.21[a] | 19.32±0.23[ab] | 29.77±0.35[ab] | 68.47±0.01[ab] | 90.25±1.01 | 75.87±0.3[ab] |
| LH-AV | 4.00±0.04[a] | 7.11±0.12[ac] | 18.55±0.19 [ab] | 26.56±0.13[ab] | 56.22±0.15[ab]c | 88.77±0.2 | 63.33±0.2[abc] |
| LH-TV | 5.82±0.17[a] | 10.07±0.08[a] | 29.78±0.12[a] | 40.55±0.02[a] | 86.22±0.03 | 95.33±0.28 | 90.44±0.1 |

Note: a = compare with control P<0.05; b = compare with LH-TV P <005; c = compare with LH-BV P<0.05.

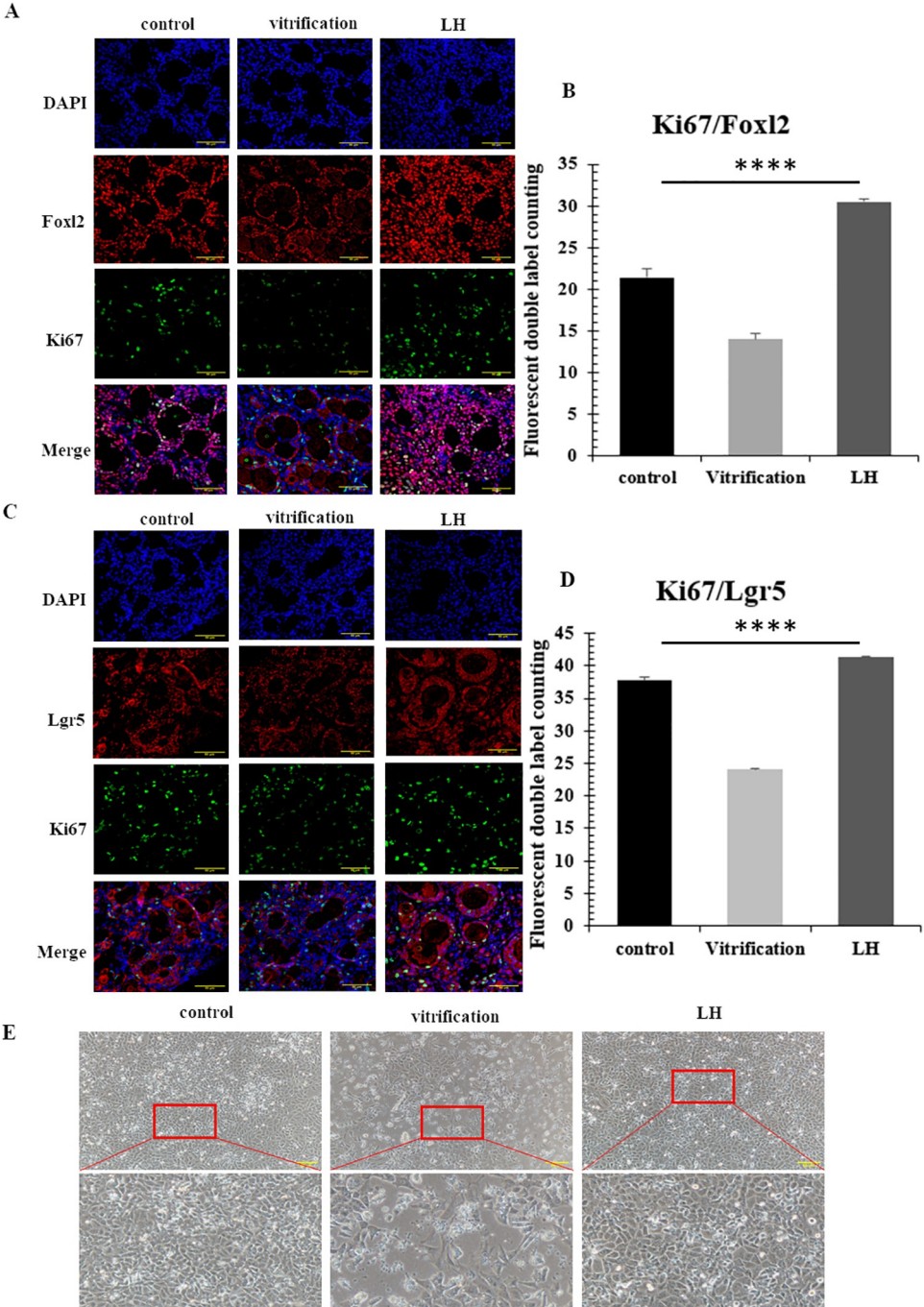

**Fig 3. Results of immunofluorescence double labeling and granulosa cell count.** (A and B) The results of double immunofluorescence assay of Ki67/Foxl2, scale = 200μm. (C and D) The results of double immunofluorescence assay of Ki67/Lgr5.scale = 200μm. (E) The results of granulosa cell culture in vitro, scale = 50μm.

## 3.3 Expression of Foxl2 and Lgr5

Immunohistochemical evaluations revealed ubiquitous expression of Foxl2 and Lgr5 in granulosa cells in all experimental groups. The LH-treated samples displayed the most robust Foxl2 expression, significantly surpassing that observed in the vitrification group, with statistical

Table 3. Granulosa cell count in 3 groups (mean ± SD) (n = 20).

| Group | Numbers of living granulosa cells | Number of dead granulosa cells | Total number of granulosa cells |
|---|---|---|---|
| control | 27.89±0.23 | 17.23±0.21 | 45.12±0.92 |
| vitrification | 10.23±1.71[ab] | 24.23±0.15[ab] | 34.46±1.78[ab] |
| LH | 39.11±0.02[a] | 14.85±1.19 | 53.96±0.03[a] |

Note: a = compare with control $P<0.05$; b = compare with LH $P<005$

significance ($P<0.05$) noted in Fig 4A and 4B. Immunofluorescence assays confirmed that Foxl2 levels were substantially elevated in the LH-treated cells compared to those in the control and vitrification groups, as demonstrated in Fig 4C and 4D ($P<0.05$). Analysis via Western blotting indicated distinct variations in Foxl2 levels among the groups, with the highest levels detected in the LH group, diminishing in the control and vitrification groups respectively, as shown in Fig 4E and 4F ($P<0.05$). Real-time PCR assays detected markedly increased Foxl2 mRNA levels in the LH group relative to the control and vitrification groups, as illustrated in Fig 4G ($P<0.05$).

Expression levels of Lgr5, akin to those of Foxl2, were significantly elevated in both LH and control groups relative to the vitrification group, with statistical significance noted ($P<0.05$, Fig 5A and 5B).Trends observed in immunofluorescence assays align with these findings, displaying the most substantial Lgr5 expression in the LH group, diminishing through the control to the vitrification groups ($P<0.05$. Fig 5C and 5D). Protein levels of Lgr5 were notably greater in the control and LH samples than in those from the vitrification group ($P<0.05$, Fig 5E and 5F). Analysis by real-time PCR indicated a significant reduction in Lgr5 mRNA levels in the vitrification samples when compared with LH and control groups ($P<0.05$, Fig 5G).

### 3.4 The result of LHR expression after LH intervention

LHR predominates on the membranes of granulosa cells. Enhanced fluorescence signaling of LHR was noted in the LH-treated group when compared to both the control and vitrification groups, with a statistical significance of $P<0.05$; the control and vitrification groups did not differ (Fig 6A and 6B). Expression levels quantified through Real-time PCR revealed that LHR mRNA was elevated in the LH group relative to the control and vitrification groups, with significance noted at $P<0.05$ (Fig 6C). The findings from Western blot analysis corroborated the immunofluorescence and mRNA expression results (Fig 6D and 6E).

### 3.5 Detection of apoptosis during vitrification cryopreservation

HE staining revealed that oocytes in the vitrification group frequently displayed nuclear contraction and assumed irregular forms. Adjacent granulosa cells were inconsistently spaced and varied in shape, leading to fragmented follicular architecture (Fig 7A). Analysis through immunohistochemistry detected localized active caspase-3 within the granulosa cells. Compared to the vitrification group, expression levels of active caspase-3 were substantially lower in both the LH group and the control, with statistical significance ($P<0.05$) (Fig 7B and 7C). This reduction was confirmed via Western blot analysis, which showed a significant decrease in active caspase-3 protein levels in the LH group ($P<0.05$) (Fig 7D and 7E). Furthermore, mRNA levels of active caspase-3 were higher in the vitrification groups than in the control and LH groups, with a significant difference ($P<0.05$) (Fig 7F).

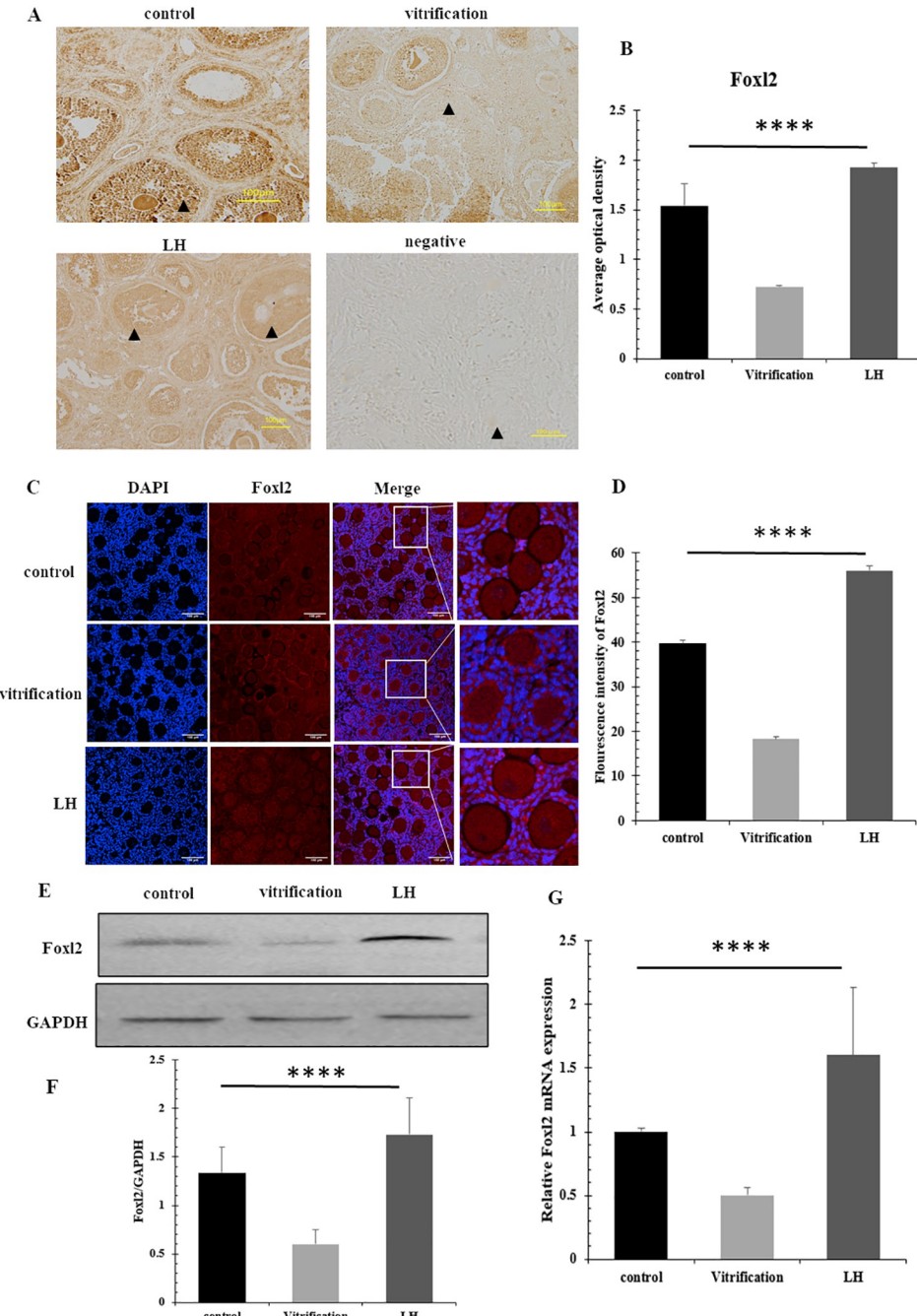

**Fig 4. The expression of Foxl2.** (A and B) Immunohistochemical detection results of Foxl2. ▲ representing follicles, scale = 100μm, (C and D) Immunofluorescence detection of Foxl2, scale = 100μm. (E and F) The expression of Foxl2 protein. (G) The mRNA levels of Foxl2.

## 3.6 Expression of TGF-β and its superfamily members

Immunohistochemical evaluations indicated a notable increase in the levels of TGF-β, GDF-9 and BMP-15 in the LH group compared with those in the vitrification group, with statistical significance ($P<0.05$) (Fig 8A–8D). Confirmation of these findings was obtained through western blot analysis (Fig 8E–8J). mRNA assessments showed significantly enhanced

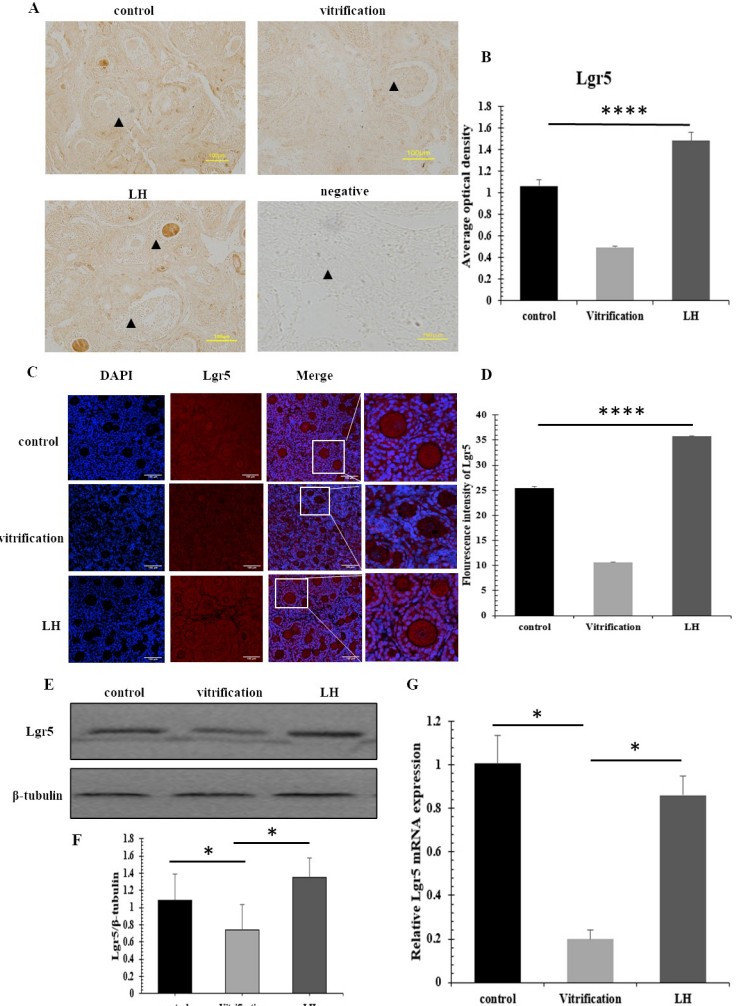

**Fig 5. The expression of Lgr5.** (A and B) Immunohistochemical detection results of Lgr5. ▲ representing follicles, scale = 100μm, (C and D) Immunofluorescence detection of Lgr5, scale = 100μm. (E and F) The expression of Lgr5 protein. (G) The mRNA levels of Lgr5.

expression of TGF-β, GDF-9, and BMP-15 in the LH group relative to the vitrification group, as evidenced by P values less than 0.05 (Fig 8K–8M).

## 4. Discussion

Reducing damage to follicles and granulosa cells during the cryopreservation process continues to pose challenges in this field. Granulosa cells modulate the function of LH, enhancing follicle development through autocrine and paracrine mechanisms [27–29]. Despite the relatively low sensitivity of LHR to LH before puberty [30], we selected 4-week-old prepubertal mice for our study. At this stage, gonadal function is relatively stable, minimizing interference from endogenous hormones and allowing the effects of exogenous hormones to be more clearly observed.

To improve the effectiveness of cryopreservation, it is crucial to investigate the optimal hormonal intervention strategy. This involves elucidating the signaling pathways that underpin the protective effects of LH on frozen-thawed follicles. The present study evaluates three LH

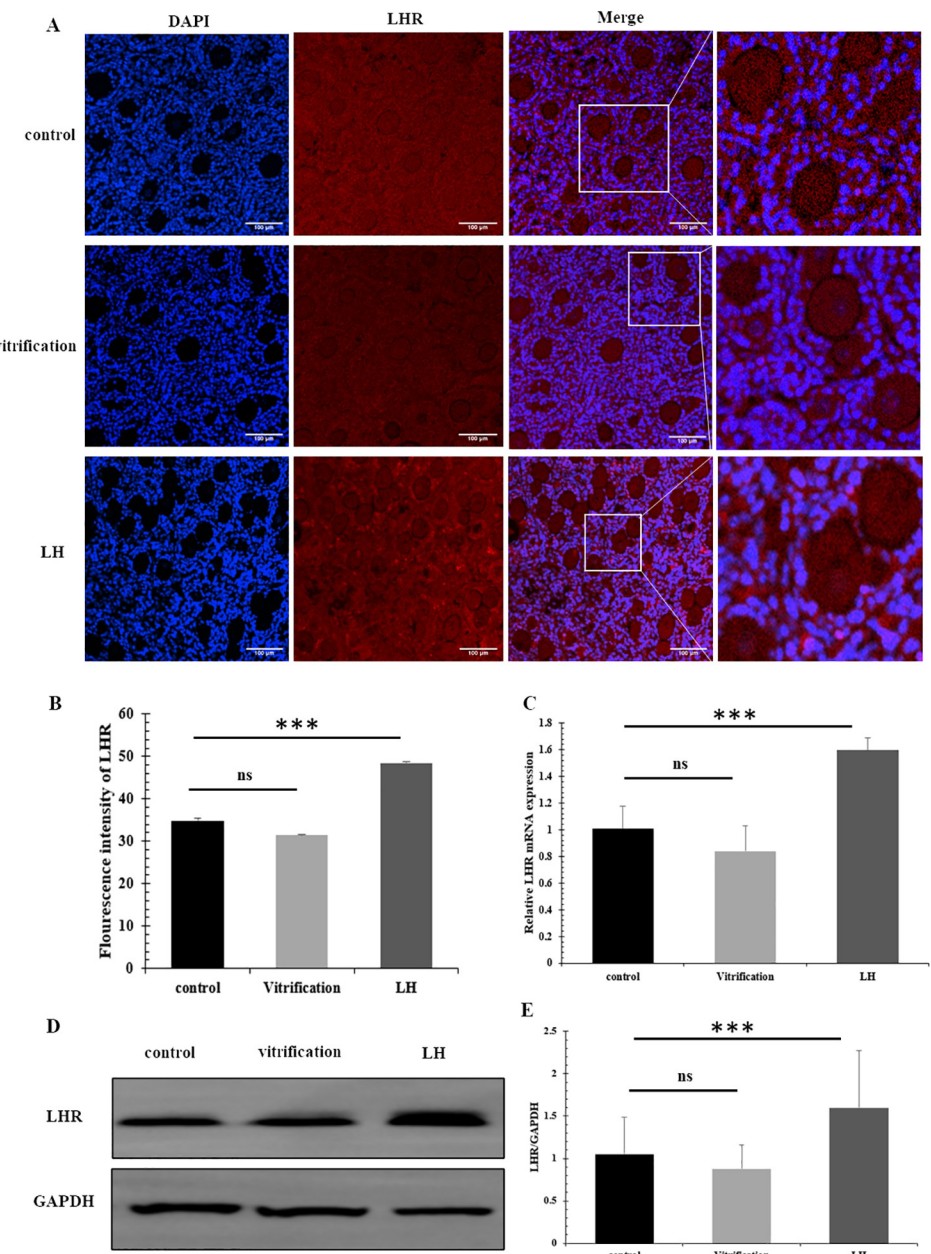

**Fig 6. The expression of Lgr5.** (A and B) Immunofluorescence detection of LHR, scale = 100μm. (C) The mRNA levels of LHR. (D and E) The expression of LHR protein.

intervention methods: before freezing, after thawing, and throughout the entire cryopreservation process. The objective is to identify the most efficacious LH strategy to enhance its protective impact and understand the associated signaling pathways.

To directly observe the impact of vitrification and cryopreservation on granulosa cells and follicular structure, all morphological assessments were conducted on cryopreserved ovarian tissue. Histological analyses using HE staining demonstrated that continuous administration of LH during vitrification and cryopreservation significantly reduced nuclear shrinkage in oocytes and apoptosis in granulosa cells, typically caused by freeze-induced damage. This

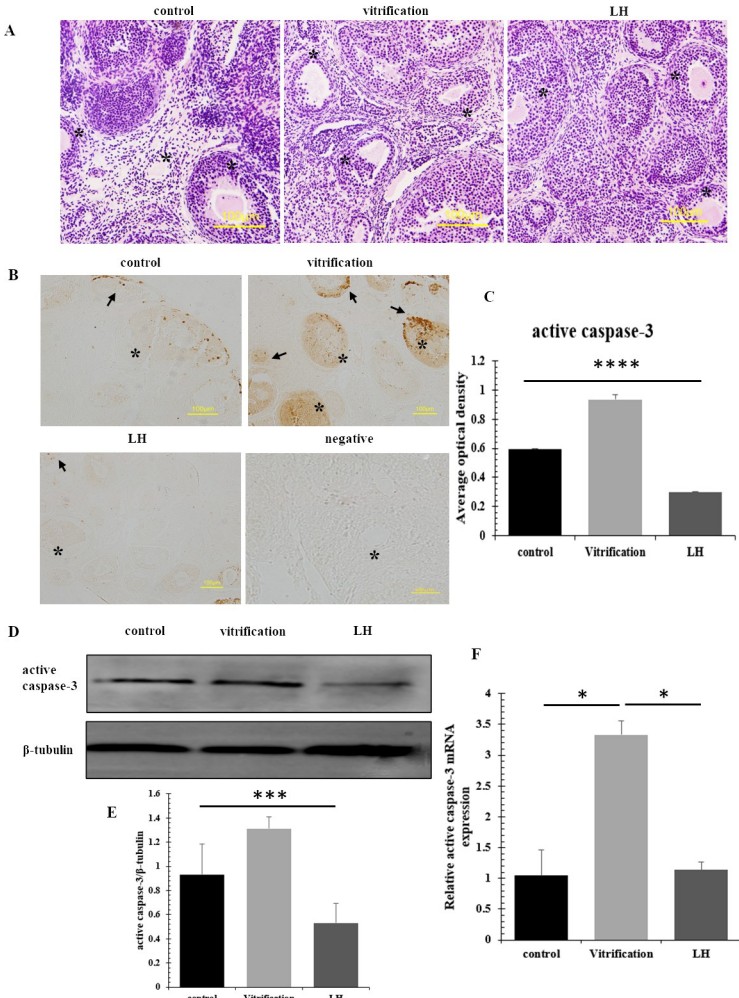

**Fig 7. Results of apoptosis detection.** (A) The morphological observation of ovaries in 3 groups, * representing follicles, scale = 100μm. (B and C) Immunohistochemical results, * representing follicles, arrow representing positive expression, scale = 100μm. (D and E) The results of western-blot. F The results of Real-time PCR.

method provided enhanced protection to follicular morphology and structural integrity compared to interventions applied only before or after cryopreservation. Endocrine function plays an important role in ovarian growth and development. Therefore, the protective regulation of hormones in the process of ovarian tissue cryopreservation is the basic guarantee for the recovery of ovarian function [31, 32]. The findings suggest that sustained LH exposure throughout the freeze-thaw cycle promotes granulosa cell proliferation and development during pre-culture and pre-equilibration phases, protects against cryodamage during vitrification, and facilitates cellular recovery during gradient thawing and subsequent culture. However, LH administration during the pre-culture and pre-equilibration phases alone offers protection only prior to freezing, failing to prevent damage during the freezing process itself. Studies have shown that the unprotected vitrification process can significantly damage the mitochondrial integrity of the granulosa cell layer and reduce the efficiency of follicle cryopreservation [33, 34]. The granulosa cells across four groups were evaluated for viability using the LIVE/DEAD assay. The proportions of live cells in both the control and LH groups were significantly higher compared to the LH-BV and LH-AV groups, under the condition of equal total cell numbers

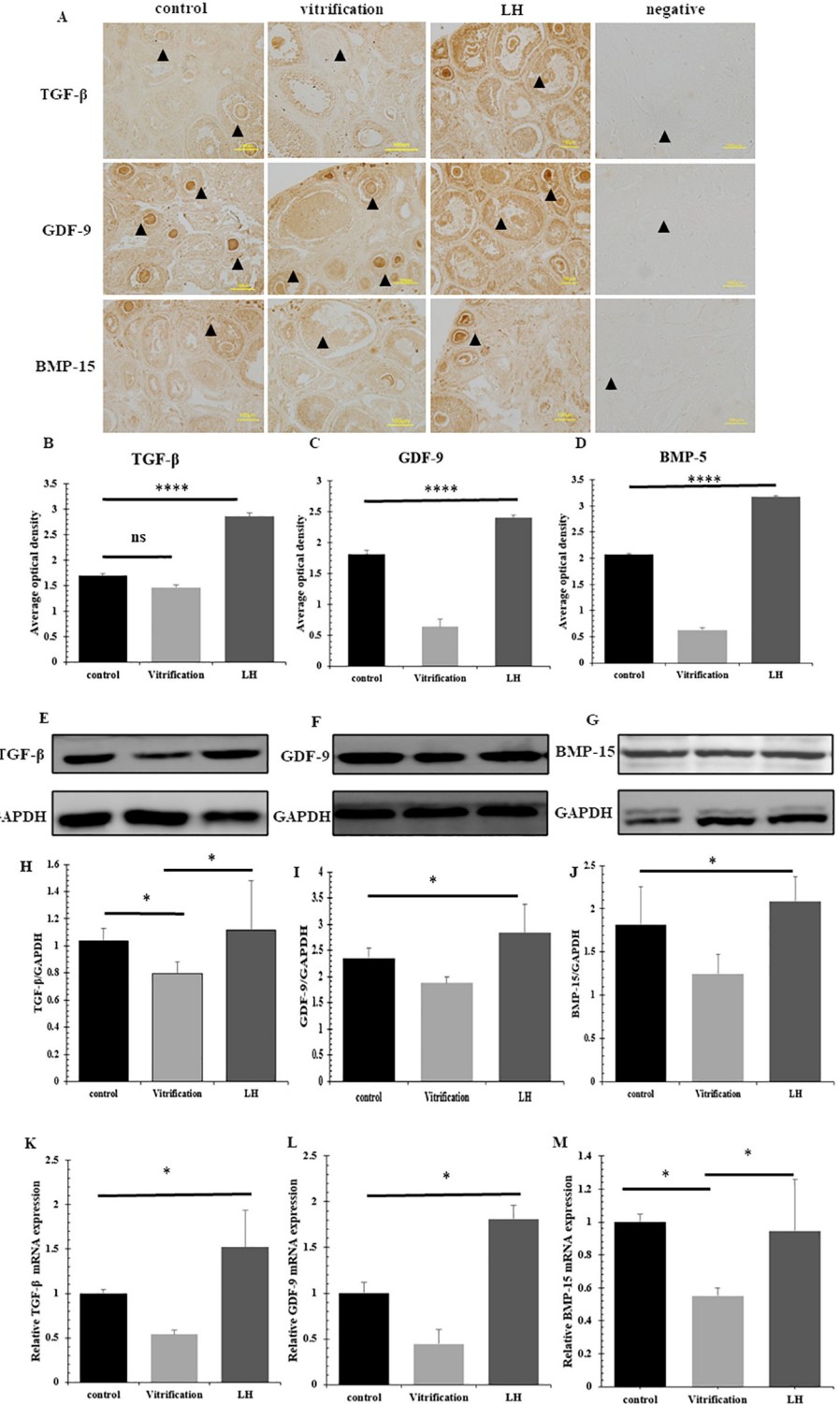

**Fig 8. The expression of TGF-β, GDF-9 and BMP-15.** (A-D) represent the localization and positive expression of TGF-β, GDF-9 and BMP-15 in the frozen-thawed ovarian tissues of the 3 groups, respectively, and ▲ represents follicles. (E-J) The protein expression of TGF-β, GDF-9 and BMP-15. (K-M) The mRNA expression of TGF-β, GDF-9 and BMP-15.

per group. This suggests that LH supplementation during the freeze-thaw process offers the most effective protection. In the assessment of viable follicles at various developmental stages, groups receiving LH treatment exhibited a significantly greater number of secondary and mature follicles than the control group. This finding aligns with the hypothesis that LHR concentration is closely associated with follicular maturity and that LHR sensitivity to LH increases with follicular volume [35]. These results reinforce earlier findings. Therefore, we believe that only LH intervention in the whole vitrification cryopreservation process is the best intervention mode to improve the efficiency of follicle cryopreservation.

We then selected the optimal LH intervention mode, which involved adding LH throughout the freeze-thaw process for the subsequent experiments. Through double immunofluorescence analysis of paraffin-embedded sections of frozen-thawed ovarian tissues across various groups, the counts of double-labeled Ki67/Foxl2 and Ki67/Lgr5 in the LH group were the highest. This indicates that the protective effect of LH maintained proliferative activity in granulosa cells post-freeze-thaw. According to literature reports, Foxl2 and Lgr5 are widely expressed in different types of granular cells [6–8], while Ki67 is mainly expressed in proliferating cells [36], so the positive expression of Ki67 is more dispersed.

In cultured granulosa cells, the vitrification group exhibited a higher number of dead cells compared to other groups, while the LH group recorded the highest number of viable cells. This indicates that LH plays a crucial role in enhancing the growth and resilience of granulosa cells against pre-freezing damage and loss during vitrification. It also aids in the functional recovery of some damaged cells, thereby reducing apoptosis after thawing. In contrast, the vitrification group, without LH intervention, displayed significantly more dead cells, suggesting that granulosa cells suffered extensive injuries leading to apoptosis due to the inability to withstand the toxic effects of the high-concentration vitrification solution. This group showed the lowest total count of granulosa cells, underscoring that without LH, the cells cannot effectively counteract apoptosis during cryopreservation, leading to inefficient vitrification and substantial cell loss.

The regulation of follicle function requires the coordinated interaction of multiple factors [37]. A critical element in this process is Foxl2, essential for the initial phases of ovarian formation and differentiation in humans. Foxl2 not only mediates the proliferation and differentiation of granulosa cells by controlling the transcription of specific genes but also enhances their proliferation by inhibiting follistatin expression [38]. Our immunohistochemical analysis revealed significantly elevated Foxl2 positive expression in the LH group compared to the control and vitrification groups. This evidence highlights the effectiveness of LH in enhancing granulosa cell growth and development. Furthermore, the levels of Foxl2 protein and mRNA in the LH group were higher than those in the other groups, consistent with previous findings that LH can significantly increase Foxl2 expression [39]. This indirectly enhances the survival conditions for granulosa cells. Concurrent detection of Lgr5, a specific marker for granulosa cell precursor development, alongside Foxl2 provides a comprehensive view of granulosa cell development and minimizes sample error. Consistent with the trend observed for Foxl2, immunohistochemistry and immunofluorescence identified the highest positive expression of Lgr5 in the LH group, although at levels lower than those of Foxl2. These results suggest that as follicles mature, Lgr5-positive granulosa cells gradually transform into Foxl2-positive cells, with Lgr5 enhancing Foxl2 expression [40]. Lgr5 protein expression was significantly higher in the LH group and markedly lower in the vitrification group, indicating Lgr5's potential role in the post-thaw repair and reconstruction of the granulosa cell growth microenvironment. The alignment of Lgr5 mRNA and protein expression underscores the effectiveness of LH intervention in increasing Lgr5-positive granulosa cells and protecting early-stage granulosa cells. This LH-induced augmentation of both Foxl2 and Lgr5 expressions ensures enhanced protection for granulosa cells derived from various sources.

Foxl2, Lgr5, and LHR are key genes expressed in human granulosa cells and form a network of mutually up-regulated genes [41]. Immunofluorescence detection indicated that LHR expression was highest in the LH group. LH binding to LHR activated its role in promoting granulosa cell development, with the addition of LH acting as a regulatory switch. This process facilitates the protection of cryopreserved granulosa cells through receptor binding, thereby enhancing follicle survival rates. The binding affinity of LH and LHR increased progressively with follicle development, aligning with the viable follicle counts observed at various developmental stages. In the LH group, both protein and mRNA levels of LHR exhibited a substantial elevation, whereas the control and vitrification groups showed no marked differences in this respect. This phenomenon can be attributed to the developmental stages of granulosa cells and follicles in the control group, which are comparatively undeveloped. At this phase, the LHR molecules present on the membranes of granulosa cells do not bind effectively with LH, resulting in limited synergistic activation between them.

The literature notes that follicle atresia commences when apoptosis rates in granulosa cells surpass 10% [42]. Subsequently, follicle membrane receptors coalesce, facilitating the receipt of apoptotic signals, triggering the apoptosis pathway, and activating previously inactive caspase-3, culminating in cell death [43]. Histological analyses using HE staining disclosed compromised morphology and structure in follicles, with an increased occurrence of apoptosis in the vitrification group. In contrast, the LH group exhibited oocytes with regular morphology and orderly granulosa cells. Immunohistochemical assays showed lower active caspase-3 expression in the LH group compared to the vitrification group, which may be linked to the effective upregulation of the granulosa cell-specific protein Foxl2 by LH. Foxl2 mitigates excessive caspase-3 activation in sinus follicles [44], boosts proliferating nuclear antigen expression in ovarian membrane and granulosa cells, and protects frozen granulosa cells by inhibiting the onset of apoptosis. The findings revealed an intriguing discrepancy: despite the differential levels of active caspase-3 protein between the control and LH groups, mRNA levels remained consistent. This observation suggests distinct regulatory mechanisms at the protein and mRNA levels. Notably, the heightened presence of active caspase-3 in the vitrification group underscores the efficacy of LH. The absence of significant mRNA alterations between the LH and control groups further supports the notion that LH offers protective benefits during cryopreservation. It appears that the cryopreservation process has minimal impact on the follicles and granulosa cells, emphasizing the critical importance of the protective role of LH. Throughout the cryopreservation, LH significantly curtailed the activation and abundance of apoptosis-related proteins in granulosa cells, thereby mitigating apoptosis.

LH is a key component of an integral endocrine regulatory pathway that synergistically interacts with TGF-β and its superfamily. This interaction critically influences granulosa cell function, indirectly promoting follicular growth and development [45]. Previous research has indicated that inhibition of the TGF-β signaling pathway significantly reduces the specific expression of LH-regulated granulosa cell proteins [46]. This underscores the close relationship between the influence of LH on oocyte and granulosa cell growth and the TGF-β signaling pathway. Immunohistochemical results showed that the LH group had the highest TGF-β positive expression compared to the control group. Although the vitrification group exhibited lower expression levels than the control group, the difference was not statistically significant. The binding affinity between LH and LHR directly impacts its function. These findings suggest that LH enhances TGF-β expression, which in turn supports the growth and development of granulosa cells [47, 48]. The significant up-regulation effect of LH on TGF-β expression was further confirmed at both protein and mRNA levels. Throughout various stages of follicular development, BMP-15 and GDF-9, members of the TGF-β superfamily, are expressed in both theca cells and granulosa cells. These proteins, characterized by homologous structures and

similar functions, are primarily secreted by oocytes [49]. The immunohistochemical localization results in this study were consistent with previous reports. Moreover, the protein and mRNA expressions of these two factors were significantly higher in the LH group, suggesting their potential cooperative role in granulosa cell growth and development. These factors likely contribute to granulosa cell growth and development via a paracrine/autocrine signaling pathway. GDF-9 primarily promotes the maturation of pre-antral follicles [50], which is consistent with the predominance of pre-antral follicles following freezing and thawing. BMP-15 not only enhances granulosa cells' sensitivity to LH but also regulates the growth and development of mouse granulosa cells by inducing and modulating LHR [6].

Following these investigations, potential interactions and connections were explored between the role of LH in enhancing the expression of specific markers in cryopreserved granulosa cells, resisting apoptosis, and the role of TGF-β and its superfamily. The findings revealed that LH promotes granulosa cell development via LHR mediation and interacts with TGF-β superfamily members BMP-15 and GDF-9 through LHR. This interaction activates the TGF-β signaling pathway, triggering anti-apoptotic processes and up-regulating the expression of specific markers in granulosa cells. Foxl2, a target gene of the TGF-β pathway, influences granulosa cell development by up-regulating genes involved in cell cycle progression and down-regulating apoptosis-related genes [51]. FOXL2 regulates gonadotropin-releasing hormone (GnRH) promoter activity and decreases GnRH receptor expression through interaction with the TGF-β signaling pathway [52]. This interaction counters GnRH-induced apoptosis. The protective effects of exogenous LH, mediated via the TGF-β signaling pathway, extend beyond granulosa cells, creating a network between granulosa cell-specific markers and the TGF-β pathway. This elucidates, from multiple perspectives, the mechanisms by which LH promotes functional recovery and apoptosis resistance in frozen-thawed granulosa cells through the TGF-β signaling pathway (Fig 9). Our research provides a foundation for

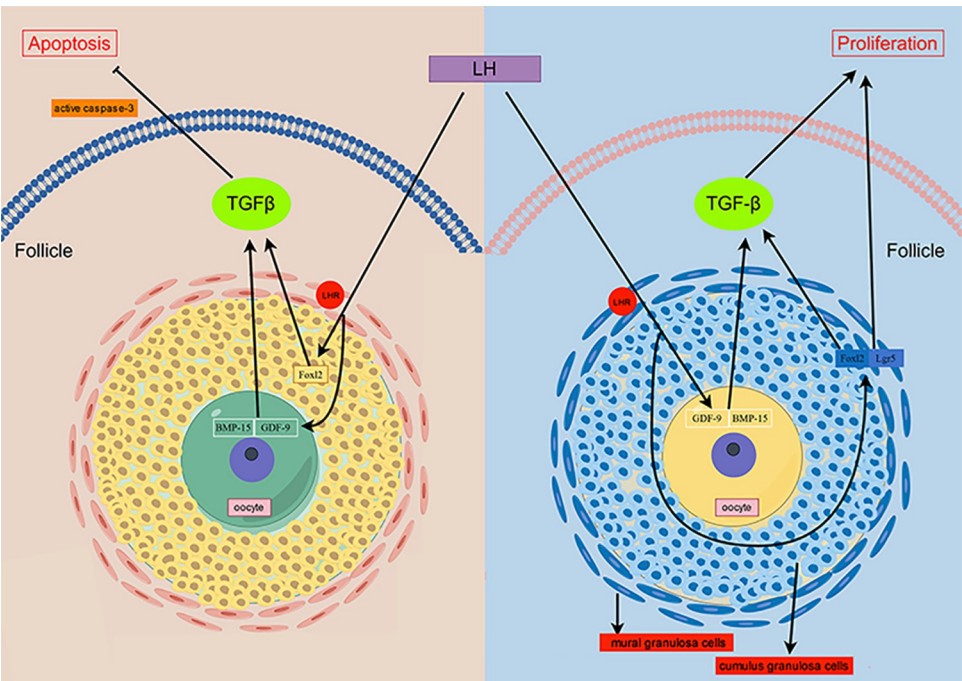

**Fig 9. The signaling pathway of LH resist apoptosis and promote proliferation of freeze-thaw folliculars and granulosa cells by TGF-β and its superfamily members.** (By figdraw).

improved protective interventions during the freeze-thaw process of follicles and granulosa cells.

## 5. Conclusion

This study proved that LH intervention in the whole process of freeze-thaw can effectively reduce the damage of follicles and granulosa cells. And the difference of TGF-β pathway in promoting follicles and granulosa cells at different developmental stages needs further study to explain.

## Supporting information

**S1 File. Raw images.**
(PDF)

**S2 File. Minimal data set.**
(PDF)

## Acknowledgments

We thanks for our laboratory members for their helpful about this study and the experimental platform support of the College of life Science, Inner Mongolia University.

## Author Contributions

**Conceptualization:** Jie Chen, Boyang Yu, Yanfeng Dai.

**Data curation:** Shengbo Zhang.

**Formal analysis:** Jie Chen.

**Funding acquisition:** Jie Chen, Boyang Yu.

**Investigation:** Boyang Yu.

**Methodology:** Shengbo Zhang, Zhikang Wang.

**Project administration:** Yanfeng Dai.

**Resources:** Yanfeng Dai.

**Supervision:** Yanfeng Dai.

**Validation:** Boyang Yu.

**Visualization:** Jie Chen.

**Writing – original draft:** Jie Chen.

**Writing – review & editing:** Jie Chen, Boyang Yu, Yanfeng Dai.

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
