## [Decision Letter · Decision Letter 0]

15 Jul 2024

PONE-D-24-22442Protective effect of luteinizing hormone on frozen-thawed ovarian follicles and granulosa cellsPLOS ONE

Dear Dr. Chen,

Thank you for submitting your manuscript to PLOS ONE. After careful consideration, we feel that it has merit but does not fully meet PLOS ONE’s publication criteria as it currently stands. Therefore, we invite you to submit a revised version of the manuscript that addresses the points raised during the review process.

We look forward to receiving your revised manuscript.

Kind regards,

Academic Editor

PLOS ONE

3. To comply with PLOS ONE submissions requirements, in your Methods section, please provide additional information regarding the experiments involving animals and ensure you have included details on (1) methods of sacrifice, (2) methods of anesthesia and/or analgesia, and (3) efforts to alleviate suffering.

 [This study was funded by the Natural and Science Foundation of Inner Mongolia Autonomous Region (No. 2019MS08139, 2022QN03003).].  

5. We note that your Data Availability Statement is currently as follows: [All relevant data are within the manuscript and its Supporting Information files.]

6. We note that Figure(s) 2, 3, 4, 5, 6, 7, and 8 in your submission contain copyrighted images. All PLOS content is published under the Creative Commons Attribution License (CC BY 4.0), which means that the manuscript, images, and Supporting Information files will be freely available online, and any third party is permitted to access, download, copy, distribute, and use these materials in any way, even commercially, with proper attribution. For more information, see our copyright guidelines: http://journals.plos.org/plosone/s/licenses-and-copyright.

a. You may seek permission from the original copyright holder of Figure(s) 2, 3, 4, 5, 6, 7, and 8 to publish the content specifically under the CC BY 4.0 license. 

7. Please include your tables as part of your main manuscript and remove the individual files. Please note that supplementary tables (should remain/ be uploaded) as separate ""supporting information"" files.

8. Please upload a copy of Supplementary Figure 1, Supplementary Figure 2, Supplementary Figure 3 and Supplementary Figure 4 to which you refer in your text on page 37 and 38. Please amend the file type to 'Supporting Information'. If the Supplementary file is no longer to be included as part of the submission please remove all reference to it within the text.

Reviewers' comments:

Reviewer's Responses to Questions

**Comments to the Author**

1. Is the manuscript technically sound, and do the data support the conclusions?

Reviewer #1: Yes

Reviewer #2: Partly

2. Has the statistical analysis been performed appropriately and rigorously? 

Reviewer #1: Yes

Reviewer #2: N/A

3. Have the authors made all data underlying the findings in their manuscript fully available?

Reviewer #1: Yes

Reviewer #2: No

4. Is the manuscript presented in an intelligible fashion and written in standard English?

Reviewer #1: Yes

Reviewer #2: No

5. Review Comments to the Author

Reviewer #1: This study found new ways in improving cryopreservation of ovarian follicles and granulosa cells where the authors reported a protective effect of Luteinizing hormone (LH). The effect involves upregulation of TGF -β and members of its superfamily like GDF-9 and BMP-15 and downregulation of active Caspase-3, the apoptosis marker.

Comments:

In Figure 2, the authors have shown results of Live/ Dead cells only for the LH TV category. Although the morphology for LH-BV and LH-AV were reported to be different from control and LH-TV cells, the Live/ Dead cells ratio for all the categories tested will help in understanding the scenario better.

Upregulation/ downregulation of Foxl2, LGR5, LHR, Caspase-3, were reported at protein level by Western Blot. Is there any specific reason why Western Blot not done for TGF -β and GDF-9 and BMP-15. (Figure 8)

Minor Comment:

The axis labels are not visible clearly for figure 4B.

Reviewer #2: The protective effect of Luteinizing hormone on frozen-thawed Ovarian follicles and Granulosa cells by Jie Chen and Yanfeng Dai et al. is a significant topic that can have substantial implications for fertility preservation, the success of assisted reproductive technologies, and the long-term reproductive health of women undergoing medical treatments that affect ovarian function. These benefits underscore the importance of continued research and application of LH in reproductive medicine.

I understand the author’s effort to make the methods easy. However, the structure and language need significant improvement; for example, the abstract could be more precise and concise. The materials and methods need proper editing to be meaningful, e.g., “Reagent preparation: thawing reagent, restore to room temperature, equipped with 1x working solution, now in use.” What do all these mean? What do LA and LB mean? “An is 2 μM”?

Materials and Methods:

2.7 does not make reading any easier. For example, “the second first antibody was added.”

2.8 the same group was mixed and randomly divided into holes.

2.9, “After allowing the cells to settle in the counting pool and fill the entire grid, counting commenced.”

These are a few examples, but they are to imply the seriousness of thorough editing.

2.14 “1ul of cDNA”: Please avoid using this kind of terminology in any scientific manuscript; instead, please add the amount of RNA used for reverse transcription.

Results

The Tiff image for Fig 2 is unavailable, and it’s too challenging to read Fig 2 B. Many of the figure labels and Tiff files are not labeled, e.g., F. I don’t know whether I'm missing something or if the journal or the authors missed it. Where are the tables in the document?

Figure 3 Authors implied that proliferative markers Ki67 co-localize with Fox12 and Lgr5. I agree with the observation, but why is the proliferation marker granular compared with the more distributed Fox12 and Lgr5? Table 3 is mentioned as missing.

The figures’ representation and clarity are commendable; however, notable errors are present, such as the overlap between Fig 4 D and B.

Figure “Real-time PCR detection of Foxl2 mRNA revealed pronouncedly higher expression in the LH group compared with in the control and vitrification groups (P<0.05) (Fig.4F). That is not true from the graph. It is almost twofold less than control. This raises the question of how protein levels are higher for Foxl2 in LH. Is there any translational regulation, protein stability, or mRNA stability? It needs an explanation from the authors. While controlled overexpression might be helpful for FoxL2 function, unregulated or excessive overexpression could harm granulosa cells and ovarian function. Please have a counterargument.

Figure 6D. The authors concluded the results: “The results of Real-time PCR detection confirmed the localization and quantitative expression of LHR, which showed that from high to low of the expression of LHR mRNA was LH group, control group, and vitrification group (P<0.05) (Fig.6D). Can a “Real-time PCR” confirm mRNA localization? The latter part is just words. Randomly put it as a sentence!

Figure 7 D & E. From the data, the authors claim that the protein levels of Caspase are different between the control and LH groups, while mRNA doesn’t have differences. How is this accounted for? The authors state that active Caspases are in mRNA and protein. Please explain how the authors inferred this from a Real-time PCR. The protein is active.

Authors should understand the expression levels of caspase are almost 3.5-fold. At the same time, Foxl2 and Lrg5 are marginal while making exuberant conclusions about the overexpression of Foxl2 and Lrg5 in the LH group compared to the control.

Figure 8. Results of functional in vitro studies involving ovarian explants are often contradictory. Several reports indicate an inhibitory effect of TGF-β1 on primary follicle survival and/or progression to the late preantral/early antral stage. However, other studies suggest positive effects or a lack of effect (Fortune 2003, Juengel & McNatty 2005). How do authors explain this?

Discussion: Many results and conclusions were made from counting and tabulating cells; unfortunately, those are unavailable for review. The tables and explanations of the above comments are required to decide on this manuscript.

6. PLOS authors have the option to publish the peer review history of their article (what does this mean?). If published, this will include your full peer review and any attached files.

Reviewer #1: No

Reviewer #2: No

---

## [Author Response · Author response to Decision Letter 0]

22 Aug 2024

Responses to the Journal’ Comments

Response: 

We sincerely appreciate the time you have dedicated to reviewing our manuscript. We are well aware of the importance of the style requirements of POLS ONE, and after carefully reading “PLOS Affiliations Formatting Guidelines” and “PLOS Manuscript Body Formatting Guidelines”, We edited the manuscript in strict accordance with the requirements. Expect to be as style requirements perfect as possible.

Response:

Thank you for your patient guidance on how to submit the original blot images and the specific details. We have carefully marked as required, and uploaded all the original image data related to the gel images in the manuscript in the form of Supporting information to ensure compliance with the requirements of the journal.

3. To comply with PLOS ONE submissions requirements, in your Methods section, please provide additional information regarding the experiments involving animals and ensure you have included details on (1) methods of sacrifice, (2) methods of anesthesia and/or analgesia, and (3) efforts to alleviate suffering. 

Response:

Thank you for your efforts to make our manuscript more compliant with journal requirements. As requested, we have added methods of sacrifice and efforts to alleviate the suffering of mice in Ethics statement of manuscript, and that methods of anesthesia and/or analgesia steps were avoided because there was no surgery or modeling in this study. The details are on page 6, lines 111-113, and marked them in red.

 [This study was funded by the Natural and Science Foundation of Inner Mongolia Autonomous Region (No. 2019MS08139, 2022QN03003).]. 

Response:

Thank you for your careful consideration of the efforts that funders put into the research. The role of the funder in this study is consistent with what the author filled out in the submission system at the time of the initial submission, as follows:

Role of Funder statement: Jie Chen: Conceptualization, Formal Analysis, Funding acquisition, Visualization, Writing - Original Draft, Writing - Review & Editing. Boyang Yu: Conceptualization, Funding acquisition, Investigation, Validation, Writing - Review & Editing.

5. We note that your Data Availability Statement is currently as follows: [All relevant data are within the manuscript and its Supporting Information files.]

Response:

Thank you for your patient guidance on the scope, content, method, etc. of submitting raw data. The original data of all experimental results involved in the manuscript have been uploaded as supporting information in a "minimum data set" PDF file, and there are no ethical or legal restrictions on all data in this paper. 

6. We note that Figure(s) 2, 3, 4, 5, 6, 7, and 8 in your submission contain copyrighted images. All PLOS content is published under the Creative Commons Attribution License (CC BY 4.0), which means that the manuscript, images, and Supporting Information files will be freely available online, and any third party is permitted to access, download, copy, distribute, and use these materials in any way, even commercially, with proper attribution. For more information, see our copyright guidelines: http://journals.plos.org/plosone/s/licenses-and-copyright.

We require you to either (1) present written permission from the copyright holder to publish these figures specifically under the CC BY 4.0 license, or (2) remove the figures from your submission: a. You may seek permission from the original copyright holder of Figure(s) 2, 3, 4, 5, 6, 7, and 8 to publish the content specifically under the CC BY 4.0 license.

We recommend that you contact the original copyright holder with the Content Permission Form (http://journals.plos.org/plosone/s/file?id=7c09/content-permission-form.pdf) and the following text:“I request permission for the open-access journal PLOS ONE to publish XXX under the Creative Commons Attribution License (CCAL) CC BY 4.0 (http://creativecommons.org/licenses/by/4.0/). Please be aware that this license allows unrestricted use and distribution, even commercially, by third parties. Please reply and provide explicit written permission to publish XXX under a CC BY license and complete the attached form.”

Response:

The strict copyright requirements of journals are the precursor guarantee for the publication of the manuscript. Thank you for your guidance in this regard. According to the requirements of the journal, we decided to delete the copyright images in the figure(s) 2, 3, 4, 5, 6, 7, and 8. At the same time, we used the original data to re-create the corresponding histogram of each figure in the Excel. We believe that the above methods can avoid copyright problems, but if our modifications are still not perfect, please indicate that we will further correct.

7.Please include your tables as part of your main manuscript and remove the individual files. Please note that supplementary tables (should remain/ be uploaded) as separate "supporting information" files.

Response:

Thank you for your precise guidance on how to insert the table as the requirments of PLOS ONE. We have placed the tables included directly after the paragraph in which they are first cited, while deleting the individual file where the tables were written. It is believed that inserting the form in this way meets the requirements of the journal.

8. Please upload a copy of Supplementary Figure 1, Supplementary Figure 2, Supplementary Figure 3 and Supplementary Figure 4 to which you refer in your text on page 37 and 38. Please amend the file type to 'Supporting Information'. If the Supplementary file is no longer to be included as part of the submission please remove all reference to it within the text.

Response:

With your patient guidance, We have removed the text on pages 37 and 38 of the original manuscript "Supplementary Figure 1”, “Supplementary Figure 2”, “Supplementary Figure 3” and “Supplementary Figure 4 ". At the same time, according to the requirements of the journal, all the original data of the blot pictures have been uploaded in the form of supporting information.

Responses to the Reviewers’ Comments

Responses to the Comments of Reviewer 1:

1. In Figure 2, the authors have shown results of Live/ Dead cells only for the LH TV category. Although the morphology for LH-BV and LH-AV were reported to be different from control and LH-TV cells, the Live/ Dead cells ratio for all the categories tested will help in understanding the scenario better.

Response: 

We sincerely thank you for taking the time to review the manuscript and for giving us the greatest encouragement and very important suggestions. Obviously, the Live/ Dead cells ratio for all the categories tested will help in understanding the scenario better. We took your advice and added Live/Dead detection experiments of all groups and the results have been supplemented in the Fig 2 B-F of munuscript. At the same time, we also have stronger evidence for the experimental results.

2.Upregulation/ downregulation of Foxl2, LGR5, LHR, Caspase-3, were reported at protein level by Western Blot. Is there any specific reason why Western Blot not done for TGF -β and GDF-9 and BMP-15. (Figure 8)

Response: 

Thank you very much for your careful review of this part of the experiment. Although we have completed the protein imprinting detection of granular cell specific expression factors, we still ignore the importance of protein detection in the aspect of signaling pathway. Therefore, according to your comments, we carefully supplemented the western blot detection of TGF-β, GDF-9 and BMP-15 in Fig 8 E-J of the manuscript. We ensured that the experimental methods and steps were consistent with the previous experiments, at the same time, the expression of target gene was further verified.

Minor Comment:

The axis labels are not visible clearly for figure 4B.

Response: 

Thank you for your accurate feedback on Figure 4B, and we apologize for the serious problems caused by the typographical error. At your suggestion, we have edited Fig 4 B to make sure everything is clearly visible.

Responses to the Comments of Reviewer 2:

 Thank you very much for your careful review and kind comments on our manuscript. Although we wrote the paper very carefully and repeatedly adjusted the language, there are still many mistakes in our expression and language structure that need to be corrected. Your comments definitely help us improve the quality of this manuscript. We have revised it point-by-point according to your comments as follows.

1. I understand the author’s effort to make the methods easy. However, the structure and language need significant improvement; for example, the abstract could be more precise and concise. The materials and methods need proper editing to be meaningful, e.g., “Reagent preparation: thawing reagent, restore to room temperature, equipped with 1x working solution, now in use.” What do all these mean? What do LA and LB mean? “An is 2 μM”? 

Response: 

We appreciate your valuable reminder to comments on the structure of the language, and we apologize for not precise and concise enough of the abstract and for the inaccurate language in the material methodology. 

In our revised manuscript, we have made specific modifications to the above problems and marked them in red, The details are as follows: 

①We rewrote the abstract to make the description as precise and concise as possible. 

②The materials and methods section has been re-edited, we have replaced the “Reagent preparation: thawing reagent, restore to room temperature, equipped with 1x working solution, now in use.” with “The process involved thawing the necessary reagents, followed by vigorous vortexing to mix 30 µL of 1.5 mM PI with 5 µL of 4mM Calcein AM with in a 10 mL solution of PBS, obtained from Seven, Beijing, China.” in the line of 169-171 of the revised munuscript.

③“LA, LB and 2 μ M ” in the original manuscript has been deleted in the revised munuscript.

Materials and Methods:

2.7 does not make reading any easier. For example, “the second first antibody was added.”

Response: 

In 2.7 of the the revised manuscript, we have replaced the“the second first antibody was added.” with “A second round of antigen retrieval preceded the addition of either rabbit-derived monoclonal antibodies against Foxl2 (1:200, abcam, UK) or Lgr5 (1:1000, abcam, UK), which were also incubated overnight at 4 °C.” in the line of 187-189 on page 9 of the revise manuscript.

2.8 the same group was mixed and randomly divided into holes.

Response: 

“the same group was mixed and randomly divided into holes.” in 2.8 in the original manuscript has been deleted in the revised manuscript.

2.9, “After allowing the cells to settle in the counting pool and fill the entire grid, counting commenced.”

These are a few examples, but they are to imply the seriousness of thorough editing.

Response: 

“After allowing the cells to settle in the counting pool and fill the entire grid, counting commenced.” in 2.9 has been deleted in revised manuscript.

In addition to the above specific errors you raised, we have also tried to correct other misrepresentations in the entire article, such as: we have replaced the “In the first experiment” in line 140 on page 7 of the original manuscript with “In the study examining optimal LH intervention protocols” in line 126 on page 6 of the revised manuscript. In addition, we have replaced the "1:0" in line 259 on page 12 of the original manuscript with "1:200" in line 233 on page 11 of the revised manuscript. And the above changes are highlighted in red. We have tried our best to correct such error

---

## [Decision Letter · Decision Letter 1]

13 Nov 2024

PONE-D-24-22442R1Protective effect of luteinizing hormone on frozen-thawed ovarian follicles and granulosa cellsPLOS ONE

Dear Dr. Chen,

Thank you for submitting your manuscript to PLOS ONE. After careful consideration, we feel that it has merit but does not fully meet PLOS ONE’s publication criteria as it currently stands. Therefore, we invite you to submit a revised version of the manuscript that addresses the points raised during the review process.

There were a few minor concerns from Reviewers that need to be fixed before we reach a final decision.

We look forward to receiving your revised manuscript.

Kind regards,

Birendra Mishra, DVM, PhD

Academic Editor

PLOS ONE

Journal Requirements:

Additional Editor Comments:

There were a few minor concerns from Reviewers that need to be fixed before we reach a final decision.

Reviewers' comments:

Reviewer's Responses to Questions

**Comments to the Author**

1. If the authors have adequately addressed your comments raised in a previous round of review and you feel that this manuscript is now acceptable for publication, you may indicate that here to bypass the “Comments to the Author” section, enter your conflict of interest statement in the “Confidential to Editor” section, and submit your "Accept" recommendation.

Reviewer #1: All comments have been addressed

Reviewer #2: All comments have been addressed

2. Is the manuscript technically sound, and do the data support the conclusions?

Reviewer #1: Yes

Reviewer #2: Yes

3. Has the statistical analysis been performed appropriately and rigorously? 

Reviewer #1: Yes

Reviewer #2: Yes

4. Have the authors made all data underlying the findings in their manuscript fully available?

Reviewer #1: Yes

Reviewer #2: Yes

5. Is the manuscript presented in an intelligible fashion and written in standard English?

Reviewer #1: Yes

Reviewer #2: Yes

6. Review Comments to the Author

Reviewer #1: Thank you for all the changes you made based on earlier review.

As the Fig 2 indicate LH is having protective effect only in the TV group w.r.t control whereas morphology differs and cell viability decreases w.r.t control in both LH-AV and LH BV groups, the authors can discuss whether it is not the effect of LH in general but only when LH is present throughout (TV) that makes that difference.

Reviewer #2: 1ul of cDNA”in 2.14in the original manuscript has been deleted.And“3 μL RNA used for reverse transcription”has been added in 2.13 in the revised manuscript line 261. Please change it to amount not the volume used

7. PLOS authors have the option to publish the peer review history of their article (what does this mean?). If published, this will include your full peer review and any attached files.

Reviewer #1: No

Reviewer #2: **Yes: **vinesh vinayachandran

---

## [Author Response · Author response to Decision Letter 1]

8 Dec 2024

Responses to the Journal’ Comments

Response: 

Thank you for your patient guidance on how to check the list of references. According to the journal requirements, we checked all the references article by article, and we confirmed that the references were complete and correct. At the same time, none of the 52 references we cited have been retracted. Among the 52 references, 4 have been added according to the comments of the reviewers. For this reason, starting from the 31st reference, the serial number of the references has been modified accordingly and marked in red.

Responses to the Reviewers’ Comments

Responses to the Comments of Reviewer 1:

1. As the Fig 2 indicate LH is having protective effect only in the TV group w.r.t control whereas morphology differs and cell viability decreases w.r.t control in both LH-AV and LH BV groups, the authors can discuss whether it is not the effect of LH in general but only when LH is present throughout (TV) that makes that difference.

Response: 

As for the supplementary discussion: "The intervention effect of LH in the whole process of vitrification and cryopreservation is significantly better than that of LH before or after vitrification cryopreservation", we thank you for your valuable reminder. We apologize for neglecting to discuss this result in the previous manuscript.

In our revised manuscript, we have included thorough discussions about these results in depth in the third paragraph of our discussion. Our explanations are primarily based on the protective effect of hormone regulation on follicles and the correlation between granulosa cells and follicle development. Detailed discussions have been added on pages 22-23, lines 432-435, 441-443, 453-455, and marked in red. We have taken precautions to ensure that the revised explanations are scientifically sound and well-supported by the available data.

Our revisions are quoted below for your convenience:

To directly observe the impact of vitrification and cryopreservation on granulosa cells and follicular structure, all morphological assessments were conducted on cryopreserved ovarian tissue. Histological analyses using HE staining demonstrated that continuous administration of LH during vitrification and cryopreservation significantly reduced nuclear shrinkage in oocytes and apoptosis in granulosa cells, typically caused by freeze-induced damage. This method provided enhanced protection to follicular morphology and structural integrity compared to interventions applied only before or after cryopreservation. Endocrine function plays an important role in ovarian growth and development. Therefore, the protective regulation of hormones in the process of ovarian tissue cryopreservation is the basic guarantee for the recovery of ovarian function [31,32]. The findings suggest that sustained LH exposure throughout the freeze-thaw cycle promotes granulosa cell proliferation and development during pre-culture and pre-equilibration phases, protects against cryodamage during vitrification, and facilitates cellular recovery during gradient thawing and subsequent culture. However, LH administration during the pre-culture and pre-equilibration phases alone offers protection only prior to freezing, failing to prevent damage during the freezing process itself. Studies have shown that the unprotected vitrification process can significantly damage the mitochondrial integrity of the granulosa cell layer and reduce the efficiency of follicle cryopreservation [33,34]. The granulosa cells across four groups were evaluated for viability using the LIVE/DEAD assay. The proportions of live cells in both the control and LH groups were significantly higher compared to the LH-BV and LH-AV groups, under the condition of equal total cell numbers per group. This suggests that LH supplementation during the freeze-thaw process offers the most effective protection. In the assessment of viable follicles at various developmental stages, groups receiving LH treatment exhibited a significantly greater number of secondary and mature follicles than the control group. This finding aligns with the hypothesis that LHR concentration is closely associated with follicular maturity and that LHR sensitivity to LH increases with follicular volume [35]. These results reinforce earlier findings. Therefore, we believe that only LH intervention in the whole vitrification cryopreservation process is the best intervention mode to improve the efficiency of follicle cryopreservation.

The references 31, 32 and 33, 34 in above paragraph are newly cited and marked in red. The new references are as follows:

[31]Wang F, Tian Y, Huang L, Qin T, Ma W, Pei C, et al. Roles of follicle stimulating hormone and sphingosine 1-phosphate co-administered in the process in mouse ovarian vitrification and transplantation. J Ovarian Res. 2023 Aug 24;16(1):173. doi: 10.1186/s13048-023-01206-1. 

[32]Bromer JG, Patrizio P. Fertility preservation: the rationale for cryopreservation of the whole ovary. Semin Reprod Med. 2009 Nov;27(6):465-71. doi: 10.1055/s-0029-1241056. 

[33]Fransolet M, Noël L, Henry L, Labied S, Blacher S, Nisolle M, et al. Evaluation of Z-VAD-FMK as an anti-apoptotic drug to prevent granulosa cell apoptosis and follicular death after human ovarian tissue transplantation. J Assist Reprod Genet. 2019 Feb;36(2):349-359. doi: 10.1007/s10815-018-1353-8.

[34]Godoy LC, Streit DP Jr, Zampolla T, Bos-Mikich A, Zhang T. A study on the vitrification of stage III zebrafish (Danio rerio) ovarian follicles. Cryobiology. 2013 Dec;67(3):347-354. doi: 10.1016/j.cryobiol.2013.10.002.Because of the insertion of new references in the discussion section, we have made corresponding changes to the serial number of references appearing in the paper, and marked them in red.

Responses to the Comments of Reviewer 2:

1. “1ul of cDNA”in 2.14in the original manuscript has been deleted.And“3 μL RNA used for reverse transcription”has been added in 2.13 in the revised manuscript line 261. Please change it to amount not the volume used.

Response: 

Thank you very much for your careful review on our manuscript. At your suggestion, we have changed "3 μL of RNA" in line 261 of manuscript 2.13 to "1 μg of RNA " and marked in red.

---

## [Editor Report · Decision Letter 2]

29 Dec 2024

Protective effect of luteinizing hormone on frozen-thawed ovarian follicles and granulosa cells

PONE-D-24-22442R2

Dear Dr. Chen,

We’re pleased to inform you that your manuscript has been judged scientifically suitable for publication and will be formally accepted for publication once it meets all outstanding technical requirements.

Kind regards,

Birendra Mishra, DVM, PhD

Academic Editor

PLOS ONE

Additional Editor Comments (optional):

Authors responded to reviewers and editor's comments.
---

## [Editor Report · Acceptance letter]

3 Jan 2025

PONE-D-24-22442R2 

PLOS ONE

Dear Dr. Chen, 

I'm pleased to inform you that your manuscript has been deemed suitable for publication in PLOS ONE. Congratulations! Your manuscript is now being handed over to our production team.

Kind regards, 

on behalf of

Dr. Birendra Mishra 

Academic Editor

PLOS ONE